# Cytokines and Chemokines in Chikungunya Virus Infection: Protection or Induction of Pathology

**DOI:** 10.3390/pathogens9060415

**Published:** 2020-05-27

**Authors:** Chintana Chirathaworn, Jira Chansaenroj, Yong Poovorawan

**Affiliations:** 1Department of Microbiology, Faculty of Medicine, Chulalongkorn University, Bangkok 10330, Thailand; chintana.ch@chula.ac.th; 2Center of Excellence in Clinical Virology, Department of Pediatrics, Faculty of Medicine, Chulalongkorn University, Bangkok 10330, Thailand; job151@hotmail.com

**Keywords:** chikungunya, cytokine, immune response

## Abstract

Chikungunya virus (CHIKV) infection has been commonly detected in tropical countries. The clinical manifestations of CHIKV infection are similar to those of rheumatoid arthritis. Outbreaks of CHIKV infection in Thailand have been reported, and the inductions of various cytokines and chemokines in CHIKV patients during those outbreaks have been shown. Although immune responses in CHIKV infection have been increasingly reported, the mechanisms associated with pathology induction are still not clearly understood. This review focuses on cytokine and chemokine production in CHIKV infection, in association with the severity of joint inflammation. Several cytokines and chemokines involved in the induction or regulation of inflammatory responses were shown to associate with the severe and persistent symptoms in CHIKV infection. Further studies on the difference in immune responses observed in an autoimmune disease, rheumatoid arthritis, infectious disease, and CHIKV infection, would provide additional insights useful for proper CHIKV therapy, especially in patients with severe joint pains.

## 1. Introduction

Chikungunya virus (CHIKV) is an arthropod-borne alphavirus of the *Togaviridae* family. It can be transmitted to humans by *Aedes aegypti* and *Aedes albopictus* mosquitoes. CHIKV infection was first identified in 1952, and has re-emerged over the last two decades. Currently, there are three genotypes with distinct antigenic characters: West African, East/Central/South African (ECSA) and Asian genotypes [1]. In 2005, a large-scale outbreak of ECSA genotype with point mutation on the E1 gene (A226V) occurred on the island of Réunion and India in 2006–2007 [2,3,4], and in 2008–2009 a large outbreak of CHIKV infection occurred in Southeast Asia [5]. Then, in 2013–2014, CHIKV was introduced to the Latin Americas [6]. Since then, CHIKV re-emerged in India with the ECSA (226A) strain in 2015, in Pakistan and Bangladesh in 2016–2017 [7,8,9] and in Thailand in 2018–2019 [10]. 

CHIKV infection is not life-threatening, and the symptoms may include fever, headache, muscle pain, joint swelling, and rash. Most patients experience swelling and joint pain, and symptoms can last for months or years. There is still no licensed vaccine or antiviral drug against CHIKV available. The molecular mechanisms underlying pathologies observed in CHIKV infection are not clearly understood, and it has not yet been established whether CHIKV can induce arthralgia directly. However, the clinical features of patients with CHIKV infection are similar to those of patients with rheumatoid arthritis (RA). It has been suggested that immune responses to CHIKV infection may promote the symptoms observed in patients with severe joint pain [11,12,13]. Further, it has been commonly reported that cytokine induction and regulation support the roles of immune responses in immunopathogenesis. Various cytokines, such as Tumor necrotic factor (TNF)-α, Interleukin (IL)-1, IL-6 and IL-18 are involved in the induction of inflammatory responses. Chemokines such as IL-8, monocyte chemo-attractant protein-1 (MCP-1) and regulated upon activation, normal T-cell expressed, and presumably secreted (RANTES) are involved in cellular recruitment. Chronic inflammatory processes could lead to the chronic pain and tissue damages. Cytokines involving in regulation of inflammatory processes, such as IL-10 and transforming growth factor (TGF)-β, play roles in controlling the inflammatory processes. For those reasons, cytokines and chemokines are commonly investigated in diseases in which inflammatory processes are involved. This brief review summarizes the studies conducted on CHIKV immune responses, with a focus on pathology induction.

## 2. Secreted Cytokines and Chemokines Induced by CHIKV Infection

Our group reported an outbreak of CHIKV infection in the southern part of Thailand in 2008 [14]. Most patients presented with typical symptoms involving severe joint pain. Sera were confirmed by the presence of immunoglobulin M (IgM) antibodies, and/or real time-polymerase chain reaction (RT-PCR) positive for CHIKV infection. Cytokines and chemokines levels in these CHIKV infection patients were investigated. Two samples, acute and convalescent sera, were collected from each patient, which meant that changes of cytokine levels could be compared in our studies. We demonstrated an induction of an interferon-inducing cytokine, IL-18, in CHIKV infection [15]. Moreover, IL-6, granulocyte colony-stimulating factor (G-CSF), granulocyte-macrophage colony-stimulating factor (GM-CSF), MCP-1, and TNF-α levels were significantly increased in acute sera, compared with the levels in control sera. The levels of these cytokines were lower in convalescent sera. In these studies, acute sera were collected 2–6 days after the onset of fever, and convalescent sera were obtained 5–13 days after retrieval of the acute sera [16]. 

## 3. Secreted Cytokines and Chemokines and Severity of CHIKV Infection 

To investigate the roles of immune responses in the severity of CHIKV infection, several studies were conducted on patients with different levels of disease severity. Additionally, another study, performed in the southern part of Thailand, investigated the role of cytokines in CHIKV infection. This study divided subjects into non-CHIKV, mild, and severe groups. Blood samples were collected on the day of presentation and 30 days later. This study showed that IL-6 levels were higher in patients with severe symptoms than in the mild symptom group. However, the severe group had significantly lower IL-8 levels than the CHIKV patients with mild symptoms [17].

Venugopalan et al. [18] reported on their survey in India. Sera collected from patients within 1 month of onset of illness were divided into three phases, acute, sub-acute, and extended symptomatic illness phases. Interferon (IFN)-α, IFN-β, IFN-γ, C-X-C motif chemokine 10 (CXCL10 also known as interferon gamma-induced protein 10, IP-10), and IL-1β were strongly induced in the early acute phase. TNF-α, MCP-1, IL-4, IL-6, and IL-10 levels were maximal in the symptomatic phase, and these maximal levels were maintained in the recovery phase. An association between cytokine levels and disease severity has also been reported, and that an increase in IL-1β and IL-6 and a decrease in RANTES protein secretion are associated with the severity of CHIKV infection [19].

Cytokine profiles were compared during acute CHIKV infection in patients with chronic joint pain (a median of 20 months after infection) and age-and gender-matched patients without joint pain. Strong cytokine responses during acute infection correlated with a decreased incidence of chronic joint pain. Low TNF-α, IL-2, IL-4, and IL-13 levels during acute infection were suggested to be predictive markers for chronic joint pain. This study determined the level of cytokines present in large sample groups (n = 121 in each group). In addition, patients with and without joint pain were compared. This report suggested that cytokine production during the acute phase facilitates viral clearance. In addition, cytokines such as IL-4 and IL-13 produced early could prevent immune-mediated arthritis [20]. 

Another study determined cytokine and chemokine profiles during the acute phase (mean 2.97 ± 1.27 d after illness onset). The results were compared with those for healthy controls. Levels of IFN-α, IL-6, IL-8, IL-10, IFN-γ, monokine induced by gamma interferon (MIG), MCP-1 and IP-10 were significantly increased compared with controls [21]. IL-17A, IL-21, IL-22, IL-27, IL-29, and transforming growth factor-beta were measured in CHIKV patients and healthy controls. IL-17A, IL-27, and IL-29 were high in patients with CHIKV infection. IL-27 levels were higher in CHIKV patients with chronic symptoms than in patients in the acute or sub-acute stage. IL-27 was significantly correlated with tender joint counts. IL-17A showed association with swollen joint counts. Interestingly, no differences in IL-21 and IL-22 levels were found between CHIKV patients and healthy controls [22]. Additionally, an association was found between cytokine levels and persistent arthralgia. This study determined cytokine and chemokine levels in healthy controls, patients with persistent arthralgia and fully recovered patients. Levels of TNF-α, matrix metalloproteinase (MMP)-1 and MMP-3 were significantly increased in patients with persistent arthralgia compared with those in fully recovered patients. A greater reduction in RANTES levels was observed in patients with severe pain than in patients with non-severe pain, whereas the increases in IFN-γ, IL-1β, IL-6, and IL-8 levels were more marked in patients with severe pain [23]. 

## 4. Secreted Cytokines and Chemokines in CHIKV Infection with Rare Clinical Manifestations

Neurological symptoms were not commonly observed in CHIKV-infected patients. Cytokines in CHIKV-infected patients with neurological symptoms have been reported [24]. In this study, patients were grouped in CHIKV infection with neurological symptoms according to their presentations with neurological syndromes, such as encephalitis, myelopathy, peripheral neuropathy, myeloneuropathy, and myopathy. Samples from eight CHIKV-infected patients were investigated in this study. Five of eight patients had neurological complications (neuro-CHIKV infected cases). Three of eight CHIKV infected patients had no sign of neurological problems (non-neuro-CHIKV infected cases). TNF-α, IL-1β, IL-6, IL-8, IL-12, IL-17A, MCP-1, RANTES, IP-10, MIG, thymus- and activation-regulated chemokine (TARC), and IFN-α levels were estimated in serum and cerebrospinal fluid (CSF) samples. In serum samples, IL-1β, IL-8, IL-17A, MCP-1, RANTES, IP-10, and TARC were higher in CHIKV infected patients than in non-CHIKV infected patients. The serum samples from the CHIKV infected patients exhibited lower levels of TNF-α, IL-6, IL-12, MIG, and IFN-α than those from the non-CHIKV-infected patients. In the neuro-CHIKV-infected cases, the expression levels of cytokines in CSF, such as TNF-α, IL-6, IL-8, MCP-1, RANTES, MIG, TARC, and IFN-α were significantly elevated, compared with those of patients with neurological symptoms due to other conditions (neuro-non-CHIKV-infected cases). IL-1β, IL-8, IL-17A, MCP-1, RANTES, IP-10, and TARC levels were elevated in the serum samples of the non-neuro-CHIKV infected cases. TNF-α, IFN-α, IL-6, IL-8, MCP-1, RANTES, MIG, and TARC levels were elevated in the CSF samples of the neuro-CHIKV infected cases. Moreover, IL-6 and IL-8 cytokine levels were elevated in the CSF samples, compared with those in the paired serum samples of the neuro-CHIKV-infected cases. Labadie et al. [25] demonstrated the presence of CHIKV antigen in spinal meninges of CHIKV infected macaques. A CHIKV antigen was found in the endothelial cells lining blood capillaries. This finding supported the induction of cytokines in CSF.

In a study by Colavita et al. [26], the cytokine profile of a lethal case of CHIKV infection was determined. IFN-α, IFN-β, and IL-6 were strongly induced in a 77-year-old CHIKV-infected patient with cardiac disease. However, in this study, the cytokines present in only one lethal case were compared with those present in four non-lethal cases.

## 5. Secreted Cytokines and Chemokines in CHIKV Infection and Viral Loads

The cytokine levels in patients with high and low viral loads were compared. Patients with high viral load had higher levels of IFN-α and IL-6 in the acute phase. Studies in animal models showed the correlation of viremia with these two cytokine levels [25,27]. IL-17 was detected in patients with persistent CHIKV infection. In addition, this study showed that high IL-6 and GM-CSF levels were associated with persistent arthralgia [28]. Another study examined the correlation between cytokine and chemokine levels and viral load. Samples collected between 2 and 10 days after the onset of symptoms were investigated. CHIKV RNA and IgM antibody were determined. In this study, patients were divided into CHIKV RNA-positive and CHIKV RNA-negative (with IgM antibody-positive) groups. IL-4, IL-5, IL-6, IL-8, IP-10, MIG, MCP-1, and RANTES levels were significantly higher in the CHIKV-infected group than in the control group. Levels of IL-4, IL-8, IP-10, MIG, MCP-1, and RANTES were higher in the CHIKV RNA-positive group than in the CHIKV RNA-negative group. Cytokine and chemokine levels in the high and low viral load groups were observed in this study. IL-6 and MCP-1 levels were significantly higher in patients with high viral loads. In addition, the cytokine levels in four patients with persistent arthralgia were followed up for 12 weeks. MIG, MCP-1, and RANTES levels gradually declined, but the levels were still much higher than in the controls. No significant differences in IL-2, IL-10, IL-12, IL-17, and TNF-α levels were found between CHIKV patients and healthy controls [29]. The association between the amount of CHIKV, as determined by RT-PCR, and the symptoms observed was also reported. Plasma samples collected during the first 4 days of illness were investigated. It is interesting to note that patients with myalgia had a significantly lower viral load than patients without myalgia [30]. 

## 6. Immunoregulation in CHIKV Infection

In addition to cytokines, components of immune regulation may play roles in the control of cytokine production. IL-18 binding protein (IL-18BP) is the secreted protein which can bind to IL-18 with high affinity. Binding of IL-18BP inhibits the IL-18 activity resulting in the reduction of IFN-γ production induced by IL18 [31]. We previously showed that the expression of IL-18BP was induced in CHIKV infection. However, IL-18BP levels have been found to be lower in convalescent sera, suggesting that IL-18BP could play a role in regulating IL-18 function in acute CHIKV infection [15]. In another study by Tripathy et al. [32], higher levels of interleukin-1 receptor antagonist—an inhibitor of IL-1α and IL-1β—were produced in patients with chronic CHIKV infection. In addition, the IL-1Ra allele 2 was found to be associated with chronic infection. This study demonstrated involvement of host genetic polymorphism in chronic CHIKV infection.

Regulatory T cells (Tregs) are cells involved in immune regulation. A lower number of Tregs is associated with autoimmune diseases. In another study, Kulkarni et al. [33] assessed the amount of Tregs and IL-10, a cytokine secreted by Tregs, in CHIKV patients. The number of Tregs was found to be lower in patients with acute and chronic CHIKV arthritis than in recovered patients or controls. IL-10 production was higher in recovered patients than in both acute and chronic patients. In addition, it has been shown that the increased number of Tregs reduced pathology in mice infected with CHIKV [34].

## 7. Conclusions

Although a number of studies have examined cytokine and chemokine in CHIKV infection, studies that investigate patients with varying levels of disease severity are necessary to confirm the role of immune responses in pathologies observed in CHIKV infection. In addition, certain cytokines are produced at certain times after CHIKV infection. The studies concerning disease severity and the kinetics of cytokine production are crucial for a detailed understanding of immunopathogenesis in CHIKV infection. Cytokines involved in inflammation and immune regulation were investigated in terms of severity of CHIKV infection. Table 1 provides a summary of studies conducted on immune responses in relation to CHIKV disease severity. The high levels of IL-6 were most suggested to associate with disease severity. IL-1β, IL-17A, IL-27, and GM-CSF levels were associated with severity of joint pain. One study showed a correlation between times of cytokine production and joint pain. It was suggested that low levels of TNF-α, IL-2, IL-4, and IL-13 during early infection were predictive markers of chronic joint pain. 

Cynomolgus macaques infected with CHIKV demonstrate similar features to human patients. It has been shown in this animal model that CHIKV antigen could be detected in various tissues up to 3 months post-infection. This suggests that the persistence of CHIKV could be another factor responsible for prolonged cytokine induction [25]. 

Rheumatoid arthritis is an autoimmune disease, in which joint and systematic inflammation is commonly observed. Several inflammatory cytokines have been shown to be influence RA severity [35]. IL-6 is a cytokine that induces various inflammatory responses, and has been shown to play a key role in RA [36]. Moreover, the efficacy of an IL-6 inhibitor and monoclonal antibodies against IL-6 receptors for RA therapy has been demonstrated. 

IL-8 is a chemokine that shows elevated levels in RA. Both IL-6 and IL-8 are major cytokines involved in joint inflammation in RA [35,37]. Gene knockout of both IL-6 and IL-8 has been found to protect against arthritis in animal models. However, no association between IL-8 levels and CHIKV infection severity has been demonstrated. In contrast, lower IL-8 levels were observed in severe-CHIKV infection groups than in mild-CHIKV infection groups [17]. In addition to IL-8, the levels of RANTES proteins were found to decrease in patients with severe pain [23]. However, elevated levels of RANTES proteins were observed in the synovial fluids, plasma, and tissues of RA patients [38,39]. IL-17, a strong inducer of IL-6 and IL-8 in RA, was found to be associated with swollen joint counts in CHIKV infection [22]. One RA study revealed that IL-7 induced the production of IL-6 and IL-8 from synovial fibroblasts [40]. Another study found that the level of IL-27, a member of the IL-6/IL-12 family, was associated with the level of RA disease activity [41], and furthermore, it was shown to correlate with tender joint counts in CHIKV infection [22]. GM-CSF activates macrophage activity, which is a crucial mechanism of RA pathogenesis. Monoclonal antibodies targeting GM-CSF receptors have, in clinical trials, been shown to be an alternative therapy for RA patients [42]. GM-CSF has been shown to be associated with persistent arthralgia in CHIKV infection [26]. The roles of the two Th2-type cytokines, IL-4, and IL-13, as anti-inflammatory cytokines have been widely demonstrated in RA [43,44]. The early production of these two cytokines has been shown to prevent immune-mediated arthritis in CHIKV infection [9]. Although several immune components have been similarly shown to be involved in joint pain in both RA and CHIKV infection, these two diseases are caused by different agents. CHIKV antigens and the predilections of this virus regarding joints are worthy of further investigation. 

In summary, reports on immune responses to CHIKV infection have suggested that certain cytokines are involved in the determining the severity of CHIKV infection. However, a larger sample size that includes samples from various time points would elucidate the role of each cytokine and the differences in data obtained from different studies. Moreover, the amount or type of CHIKV infection present in different individuals could play a role in the emergence of different clinical manifestations. Lastly, host immunogenetics could be another factor influencing the differences observed in CHIKV infection [45,46]. 

## Figures and Tables

**Table 1 pathogens-09-00415-t001:** Studies on cytokine levels in relation to severity of chikungunya virus infection.

Cytokine	Finding	Reference
IL-6	Higher in severe than in mild group	[17]
IL-8	Lower in severe than in mild symptoms	[17]
TNF-α, IL-2, IL-4, IL-13	Low level during acute infection and suggested as predictive markers for chronic joint pain	[20]
IL-4, IL-13	Early production could prevent immune-mediated arthritis	[20]
IL-27	Correlated with tender joint counts	[22]
IL-17A	Associated with swollen joint counts	[22]
TNF-α, MMP-1 MMP-3	Higher in patients with persistent arthralgia than in fully recovered patients	[23]
RANTES	Lower in patients with severe pain than in patients with non-severe pain	[23]
IFN-γ, IL-1β, IL-6, IL-8	Higher in patients with severe pain than in patients with non-severe pain	[23]
IL-6, GM-CSF	Associated with persistent arthralgia	[26]

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
