# Peer review of "Cytokines and Chemokines in Chikungunya Virus Infection: Protection or Induction of Pathology"

_pathogens, 2020, doi:10.3390/pathogens9060415_

Round 1

Reviewer 1 Report

The review paper by Chirathaworn et al. is a very nice and complete overview of immune responses following Chikungunya virus infection and how these relate to immune protection and pathology.

The authors focuss their review on human and non-human primate data, which makes it particularly relevant and compelling.

There are no major remarks or flaws in this review paper, apart from a few small typo's that can be addressed during the production stage. 

Author Response

Reviewer 1 comment

Comments and Suggestions for Authors

The review paper by Chirathaworn et al. is a very nice and complete overview of immune responses following Chikungunya virus infection and how these relate to immune protection and pathology.

The authors focuses their review on human and non-human primate data, which makes it particularly relevant and compelling.

There are no major remarks or flaws in this review paper, apart from a few small typo's that can be addressed during the production stage. 

Responses to the Reviewer 1 comment.

We would like to thank the Reviewer 1 very much for the very kind and encouraging comments.   We would like to apologize for the typo, and we have already proofread and made thecorrection.

Reviewer 2 Report

In this review, the authors summarise current literature detailing the immune responses to Chikungunya virus. This is clearly in important topic, particularly as CHIKV can lead to chronic diseases similar to conditions characterized by immunological dysregulation such as rheumatoid arthritis. 

This review is pretty limited in scope and lacks any real depth. Each section is dominated by long lists of cytokines that could more concisely be summarized with a table. Therefore, it reads more like the initial stages of a short meta-analysis The review lacks any clear structure and it is not clear to me what the take-home message or purpose of the review is. If there is an overall hypothesis or model, then this should be presented including a figure/schematic. Similarly, what the significance of the various cytokine expression could be is not discussed; i.e. what is surprising to find? what is similar to other viruses? How do the observed cytokines inform on CHIKV virology?

Author Response

Reviewer  2 comment

Comments and Suggestions for Authors

In this review, the authors summarize current literature detailing the immune responses to Chikungunya virus. This is clearly in important topic, particularly as CHIKV can lead to chronic diseases similar to conditions characterized by immunological dysregulation such as rheumatoid arthritis. 

This review is pretty limited in scope and lacks any real depth. Each section is dominated by long lists of cytokines that could more concisely be summarized with a table. Therefore, it reads more like the initial stages of a short meta-analysis.  The review lacks any clear structure and it is not clear to me what the take-home message or purpose of the review is. If there is an overall hypothesis or model, then this should be presented including a figure/schematic. Similarly, what the significance of the various cytokine expression could be is not discussed; i.e. what is surprising to find? What is similar to other viruses? How do the observed cytokines inform on CHIKV virology?

Responses to the Reviewer 2 comment.

          We would like to thank the Reviewer 2 for the comments.  Would it be possible for the Reviewer to suggest which part (s) of this review should be revised?

Reviewer 3 Report

In this article, Chintana Chirathaworn et al discussed about the immune responses against chikungunya and the impact on the disease outcome from the Thailand experience in comparison to the published works from other countries.

I suggest to intitled it: “Immune Responses in Chikungunya Virus Infection, from Protection or Induction of Pathology : the Thailand experience.”

The fact that this paper is mainly fuelled by the studies from Thailand deserved (must) to be indicated in the title AND in the abstract because this is a major advance of this work.

I have only minor comments that I fell will help to understand the final discussion conclusion.

Line 112: is there CSF samples from the patient described here ?

Line 121, a comment about the cytokine level in CSF of chikv infected macaque may help here (there are provided in Labadie et al. 2010 supl. Data).

Lne 128 concerning the acute phase, you should add reference to the animal models showing that shown a clear relationship between the vrial load and the level of IFNa and IL6 in the acute phase (Labadie et al. 2010, Roques et al. viruses 2018)

Finally the discussion is well done and not any point was missed.

However in the final chapter, line 209 please cut the word “indifferent”; to note there are some studies trying to decifer the genetic relationship and severity of the disease that deserve to be cited here. Because in my knowledge any of them were able to sustain this conclusion up to now.

I found other few minor typo correction that need to be corrected.

Author Response

Reviewer 3 comment

Comments and Suggestions for Authors

In this article, Chintana Chirathaworn et al discussed about the immune responses against chikungunya and the impact on the disease outcome from the Thailand experience in comparison to the published works from other countries.

I suggest to entitle it: “Immune Responses in Chikungunya Virus Infection, from Protection or Induction of Pathology: the Thailand experience.”

The fact that this paper is mainly fuelled by the studies from Thailand deserved (must) to be indicated in the title AND in the abstract because this is a major advance of this work.

Responses to the Reviewer 3 comments.

          I would like to thank Reviewer 3 very much for yourkind suggestions.  We are very pleased that the work done in Thailand is well recognized by the Reviewer.  This is very encouraging.  However, this is a review article including the works reported by several groups.  It could not be formulated to be a review articlethat got the reviewers’ compliments without the works published by others.   We have mentioned clearly the amount of work done in Thailand in the manuscript. We hope the Reviewer does not mind if we leave the tile as it is. 

I have only minor comments that I fell will help to understand the final discussion conclusion.

Line 112: is there CSF samples from the patient described here?

          Line 112 described cytokines in sera.  To make these more transparent, we have made some corrections, as shown in the revised manuscript.  Cytokines in CSF were mentioned later from Line 114. 

Line 121, a comment about the cytokine level in CSF of chikv infected macaque may help here (there are provided in Labadie et al. 2010 supl. Data).

          We have read the suppl data of Labadie et al. .2010 again.  We could not find the experiment on cytokines in CSF.  Table 3S showed CCL2, IL-6, CD14+ cells, and viral load in blood.  However, we have added the information about CHIKV antigen in spinal meninges as shown in the revised manuscript.

Line 128 concerning the acute phase, you should add reference to the animal models showing that shown a clear relationship between the viral load and the level of IFNa and IL6 in the acute phase (Labadie et al. 2010, Roques et al. viruses 2018)

          These two references were mentioned and added, as shown in the revised manuscript.

Finally the discussion is well done and not any point was missed.

          Thank you very much for this encouraging comment.

However, in the final chapter, line 209, please cut the word “indifferent”; to note there are some studies trying to decifer the genetic relationship and severity of the disease that deserve to be cited here. Because in my knowledge any of them were able to sustain this conclusion up to now.

          The word “indifferent” was cut, and the references for immunogenetic studies were added as suggested.

I found other few minor typo correction that need to be corrected.

          We have tried to proofread and made corrections.

Reviewer 4 Report

Pathogens-756453

Title: Immune Responses in Chikungunya Virus Infection: Protection or Induction of Pathology

Corresponding author: Yong Poovorawan

This manuscript reviewed the current literature on cytokine/chemokine expression during chikungunya virus infection in humans. Additionally, the authors described changes in cytokine/chemokine expression during various stages and severity of CHIKV disease and viral load. The authors try to compare the CHIKV findings to rheumatoid arthritis (RA), but do not provide enough information about RA in the review. Overall, the authors provide a nice review on this topic, but additional information is needed for acceptance.

Major Comments:

  1. The table is a bit complicated to look at since the same cytokine is sometimes listed more than once. The table would be more streamline if each cytokine was listed once and there were more columns to describe if there is increased or decreased expression (arrows could be used) during severe/mild disease, acute/chronic, viral burden, etc. compared to a control group. Also, remove “of studies on” from the title.
  2. The authors use “immune response” throughout the manuscript to describe cytokine and chemokine expression. While cytokines and chemokines are an aspect of the immune response, there are many other immune components (e.g. immune cells, antibodies, interferons) that are not discussed in this review. The authors should write more specific title, headings, and clarify within the text.
  3. Please include a section comparing RA and CHIKV disease rather than just in the conclusion. In this section, please include more detail as to how these diseases are similar in terms of clinical manifestation which could be related to cytokine levels.

Minor Comments:

  1. From lines 32-36, the authors introduce the A226V mutation in the E1 protein and related outbreaks. However, some of the indicated outbreaks were not with the ESCA genotype (e.g. the 2013-2014 outbreak in Latin America was the Asian genotype). Please discuss the mutations and indicate the appropriate genotype in each outbreak.
  2. Please add more information to the type of samples collected on line 144 and the relationship of cytokines and other symptoms of CHIKV infection (e.g. myalgia).
  3. Line 38: please include other symptoms of CHIKV infection and a reference
  4. Line 78-80, 87-88, 97-98, 127, 130-131, 179-180, 184-185: please include references
  5. Please include the neurological symptoms observed.
  6. Remove line 160-161. No other mouse data was included in this review.
  7. Please include more information about IL-18 binding protein in section 6. Does it inhibit IL-18 activity?
  8. Misspelling: Line 55 (granulocytecolony); line 113 (thenon); line 155 (Tregsis)

Author Response

Reviewer 4 comment

Comments and Suggestions for Authors

Pathogens-756453

Title: Immune Responses in Chikungunya Virus Infection: Protection or Induction of Pathology

Corresponding author: Yong Poovorawan

This manuscript reviewed the current literature on cytokine/chemokine expression during chikungunya virus infection in humans. Additionally, the authors described changes in cytokine/chemokine expression during various stages and severity of CHIKV disease and viral load. The authors try to compare the CHIKV findings to rheumatoid arthritis (RA), but do not provide enough information about RA in the review. Overall, the authors provide a nice review on this topic, but additional information is needed for acceptance.

Major Comments:

The table is a bit complicated to look at since the same cytokine is sometimes listed more than once. The table would be more streamline if each cytokine was listed once and there were more columns to describe if there is increased or decreased expression (arrows could be used) during severe/mild disease, acute/chronic, viral burden, etc. compared to a control group. Also, remove “of studies on” from the title.

Responses to the Reviewer 4 comment.

          We would like tothank the reviewer 4 very much for the comment about the list of cytokines in Table 1.  At first, we have tried to do as the Reviewer suggested.  However, we have found as follows:

  1. Several cytokines mentioned were increased compared with healthy individuals. The studies areshown in theTable I demonstrated the higher levels between different groups of patients, not the induction of cytokines. For that reason, we decide not to use the arrows.
  2. For some cytokines, the findings in different reports were different. In addition, some studies reported the association with severity, but others focused on persistence and recovered stages. 

For the above reasons, the same cytokine was mentioned more than once in the Table.

The authors use “immune response” throughout the manuscript to

describe cytokine and chemokine expression. While cytokines and chemokines are an aspect of the immune response, there are many other immune components (e.g. immune cells, antibodies, interferons) that are not discussed in this review. The authors should write more specific title, headings, and clarify within the text.

The title was changed, as the Reviewer suggested.

Please include a section comparing RA and CHIKV disease rather than just in the conclusion. In this section, please include more detail as to how these diseases are similar in terms of clinical manifestation which could be related to cytokine levels.

          The comparison of RA and CHIKV infection is interesting, as the Reviewer mentioned.  There are several reports and review articles on cytokine production in RA.  In our studies, in Thailand, cytokines commonly found to associate with RA were selected for studying their roles in joint inflammation in CHIKV infection.  Cytokines commonly found in both RA and CHIKV are involved in the induction of inflammatory responses and cell recruitment.  These suggest that inflammatory processes are involved in both RA and CHIKV.   In this review article, we would like to focus on the comparison among CHIKV infected patients with different disease onsets.  Information on the difference between the rheumatogenic autoantigens and CHIKV antigens in the induction of inflammatory responses could provide more information to clarify the mechanisms of pathogenesis induction in both diseases.  Certainly, the review article comparing between immune response inductions by autoimmune diseases and CHIKV infection is worth to be considered. 

Minor Comments:

From lines 32-36, the authors introduce the A226V mutation in the E1 protein and related outbreaks. However, some of the indicated outbreaks were not with the ESCA genotype (e.g. the 2013-2014 outbreak in Latin America was the Asian genotype). Please discuss the mutations and indicate the appropriate genotype in each outbreak.

          In 2005, a large-scale outbreak of ECSA genotype with point mutation on the E1 gene (A226V) occurred on the island of Réunion and India in 2006–07, and in 2008–09 a large outbreak of CHIKV infection occurred in Southeast Asia.  Then, in 2013–14, CHIKV was introduced to the Latin Americas. Since then, CHIKV re-emerged in India with the ECSA (226A) strain in 2015, in Pakistan and Bangladesh in 2016–17 and in Thailand in2018–19.

Please add more information to the type of samples collected on line 144 and the relationship of cytokines and other symptoms of CHIKV infection (e.g. myalgia).

          The type of samples has been included, as shown in the revised manuscript.  Mechanisms of pathological symptoms caused by the inflammatory processes have commonly involved the increase of vascular permeability, fluid leakage, and cellular recruitment.  These processes lead to pain, swelling, edema and redness.  These processes are caused by the functions of several cytokines and chemokines.  Chronic inflammatory responses could result in chronic pain or pathology.  Anti-inflammatory cytokines and the regulators of inflammatory responses are important in controlling the inflammatory processes.   The information about these processes has been added at the end of the introduction section. 

Line 38: please include other symptoms of CHIKV infection and a reference

          Other symptoms have been included, as shown in the revised manuscript.

Line 78-80, 87-88, 97-98, 127, 130-131, 179-180, 184-185: please include references

          The reference for Line 78-80 was included at the end of this paragraph (Chang et al. 2018). 

          The reference for Line 87-88 was shown in Line 90 (Tanabe et al 2019)

          The reference for Line 97-98 was shown in Line 102 (Ninla-Aesong et al 2019)

          The reference for Line 127 was shown in Line 130 (Chow et al 2011)

          The reference for Line 130-131 was shown in Line 142 (Reddy et al 2014)

          The reference for Line 179-180 was added(McInnes et al. 2015)

          The reference for Line 184-185 was added (Magyari et al.2014, McInnes et al. 2015)

Please include the neurological symptoms observed.

          The neurological symptoms have been included where the report of Kashyap et al. 2014 was mentioned.

Remove line 160-161. No other mouse data was included in this review.

          Although this review focuses mainly on the studies in human patients, additional information in animal models, especially on the mechanisms that are not widely studied in humans, could support findings in humans.  We hope the Reviewer does not mind if we still keep these two lines.  

Please include more information about IL-18 binding protein in section 6. Does it inhibit IL-18 activity?

           This information should have been added as Reviewer 4 suggested.   Brief information about theIL18-BP function was added, and the reference was included, as shown in the revised manuscript.

Misspelling: Line 55 (granulocytecolony); line 113 (thenon); line 155 (Tregsis)

          These occurred when the pages/paragraphs were rearranged, and we have made corrections as the Reviewer 4 suggested. 

Round 2

Reviewer 3 Report

I found the new title more accurate and most of the comments are adressed by the author

However the abstract is really poorly informative. There is no clear description, in addition even the indication that a large part of this review is supported by comparieson with the Thailande data presented by the authors must be indicated in this abstract.

Author Response

Comments and Suggestions for Authors

I found the new title more accurate and most of the comments are adressed by the author

However the abstract is really poorly informative. There is no clear description, in addition even the indication that a large part of this review is supported by comparieson with the Thailande data presented by the authors must be indicated in this abstract.

The studies of the outbreaks and cytokine/chemokines in CHIKV infection in Thailand were mentioned in the abstract as the reviewer suggested. 

Abstract

Chikungunya virus (CHIKV) infection has been commonly detected in tropical countries.  The clinical manifestations of CHIKV infection are similar to those of rheumatoid arthritis. The outbreaks of CHIKV infection in Thailand have been reported.  In addition, the inductions of various cytokines and chemokines in CHIKV patients during those outbreaks in Thailand were shown.  Although immune responses in CHIKV infection have been increasingly reported, the mechanisms associated with pathology induction are still not clearly understood.  This review focuses on cytokine/chemokine production in CHIKV infection in association with the severity of joint inflammation.  Several cytokines/chemokines involving in the induction or regulation of inflammatory responses were shown to associate with the severe and persistent symptoms in CHIKV infection.  Further studies on the difference in immune responses observed in an autoimmune disease, rheumatoid arthritis, and infectious disease, CHIKV infection, would provide additional insights useful for proper CHIKV therapy, especially in patients with severe joint pains.